# ADVERSARIAL STYLE TRANSFER FOR ROBUST POLICY OPTIMIZATION IN REINFORCEMENT LEARNING

## ABSTRACT

This paper proposes an algorithm that aims to improve generalization for reinforcement learning agents by removing overfitting to confounding features. Our approach consists of a max-min game theoretic objective. A generator transfers the style of observation during reinforcement learning. An additional goal of the generator is to perturb the observation, which maximizes the agent's probability of taking a different action. In contrast, a policy network updates its parameters to minimize the effect of such perturbations, thus staying robust while maximizing the expected future reward. Based on this setup, we propose a practical deep reinforcement learning algorithm, Adversarial Robust Policy Optimization (ARPO), to find a robust policy that generalizes to unseen environments. We evaluate our approach on Procgen and Distracting Control Suite for generalization and sample efficiency. Empirically, ARPO shows improved performance compared to a few baseline algorithms, including data augmentation.

## 1 INTRODUCTION

The reinforcement learning (RL) environments often provide observation, which is a high-dimensional projection of the true state, complicating policy learning as the deep neural network model might mistakenly correlate reward with irrelevant information. Thus deep neural networks might overfit irrelevant features in the training data due to their high flexibility and the long-stretched time duration of the RL training process, which leads to poor generalization (Hardt et al., 2016; Zhang et al., 2018a; Cobbe et al., 2019; 2020; Zhang et al., 2018b; Machado et al., 2018; Gamrian & Goldberg, 2019). These irrelevant features (e.g., background color) usually do not impact the reward; thus, an optimal agent should avoid focusing on them during policy learning. Even worse, this might lead to incorrect state representations, which prevents deep RL agents from performing well even in slightly different environments. Thus, to learn a correct state representation, the agent needs to avoid those features that are not important for optimal policy learning.

In recent times, generalization in RL has been explored extensively. In zero-shot generalization framework (Zhang et al., 2018b; Song et al., 2020; Wang et al., 2019; Packer et al., 2018) the agent is trained on a finite training set of MDP levels and then evaluated on the full distribution of the MDP levels. The distribution of the MDP levels can be generated in various ways, including changing dynamics, exploit determinism, and adding irrelevant confounders to high-dimensional observation. An agent trained in these setups can overfit any of these factors, resulting in poor generalization performance. Many approaches have been proposed to address this challenges including various data augmentation approaches such as random cropping, adding jitter in image-based observation (Cobbe et al., 2019; Laskin et al., 2020a; Raileanu et al., 2020; Kostrikov et al., 2020; Laskin et al., 2020b),random noise injection (Igl et al., 2019), network randomization (Osband et al., 2018; Burda et al., 2018; Lee et al., 2020), and regularization (Cobbe et al., 2019; Kostrikov et al., 2020; Igl et al., 2019; Wang et al., 2020) have shown to improve generalization. The common theme of these approaches is to increase diversity in the training data so as the learned policy would better generalize. However, this perturbation is primarily done in isolation of the RL reward function, which might change an essential aspect of the observation, resulting in sub-optimal policy learning. Moreover, the random perturbation in various manipulations of observations such as cropping, blocking, or combining two random images from different environment levels might result in unrealistic observations that the agent will less likely observe during testing. Thus these techniques might work poorly in the setup where agents depend on realistic observation for policy learning. It is also de-

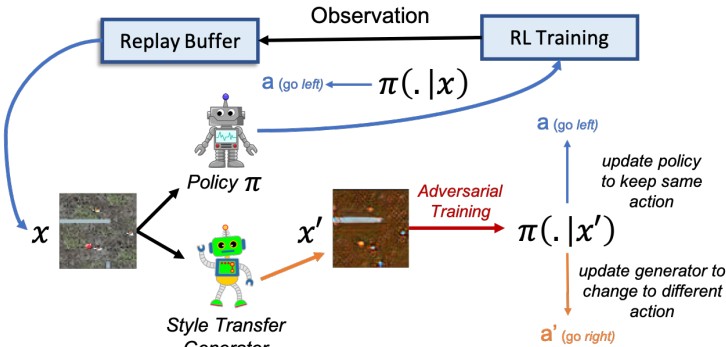

Figure 1: Overview of our approach. The method consists of a minimax game theoretic objective. It first applies a clustering approach to generate $n$ group of clusters based on the different visual features in the observation. A generator is then used to style-translate observation from one cluster to another while maximizing the change in action probabilities of the policy. In contrast, the policy network updates its policy parameters to minimize the translation effect while optimizing for the RL objective that maximizes the expected future reward.

sirable to train the agent with realistic observations, which helps it understand the environments' semantics. Otherwise, the agent might learn unexpected and unsafe behaviors while entirely focusing on maximizing rewards even by exploiting flaws in environments such as imperfect reward design.

This paper focuses on a particular form of generalization where the agent gets to observe a high-dimensional projection of the true state. In contrast, the latent state dynamics remain the same across all the MDP levels. This assumption has been shown to affect generalization (Song et al., 2020) in well studied benchmarks (Nichol et al., 2018) . It has been observed that the agent overfit the scoreboard or timer and sometimes achieves the best training performance even without looking at the other part of the observation. Consequently, this policy completely fails to generalize in test environments. The high-capacity model even can memorize the whole training environment, thus severely affecting the generalization (Zhang et al., 2018b).

This paper proposes a style transfer-based observation translation method that considers the observation's content and reward signal. In particular, a generator performs the style transfer while the policy tries to be robust toward such style changes. Note that, style transfer-based approach has been used to target generalization in Anonymous (2021) where the authors propose to train style transfer and reinforcement learning in isolation. In contrast, in this paper, we propose a unified learning approach where the style transfer is integrated into the reinforcement learning training. Thus the generator produces more effective translation targeted toward RL objective, making the RL policy more robust. The trajectory data from the agent's replay buffer is first clustered into different categories, and then observation is translated from one cluster's style to another cluster's style. Here the style is determined by the attribute commonality of observation features in a cluster. The agent should be robust toward changes of such features. Moreover, the translated trajectories correspond to those that possibly appear in testing environments, assisting the agent in adapting to unseen scenarios.

Figure 1 shows an overview of our method. Our approach consists of a max-min game theoretic objective where a generator network modifies the observation by style transferring it to maximize the agent's probability of taking a different action. In contrast, the policy network (agent) updates its parameters to minimize the effect of perturbation while maximizing the expected future reward. Based on this objective, we propose a practical deep reinforcement learning algorithm, Adversarial Robust Policy Optimization (ARPO), that aims to find a robust policy by mitigating the observational overfitting and eventually generalizes to test environments.

We empirically show the effectiveness of our ARPO agent in generalization and sample efficiency on challenging Procgen (Cobbe et al., 2020) and Distracting Control Suite (Stone et al., 2021) benchmark with image-based observation. The Procgen leverages procedural generation to create diverse

environments to test the RL agents' generalization capacity. The distracting control adds various visual distractions on top of the Deepmind Control suite (Tassa et al., 2020) to facilitate the evaluation of generalization performance.

Empirically, we observed that our agent ARPO performs better in generalization and sample efficiency than the baseline PPO (Schulman et al., 2017) and a data augmentation-based RAD (Laskin et al., 2020a) algorithm on the many Procgen environments. Furthermore, APRO performs better compared to PPO and SAC (Haarnoja et al., 2018) on the evaluated two Distracting Control Suite environments in various setups.

In summary, our contributions are listed as follows:

- We propose Adversarial Robust Policy Optimization (ARPO), a deep reinforcement learning algorithm to find a robust policy that generalizes to test environments in a zero-shot generalization framework.

- We evaluate our approach on challenging Procgen (Cobbe et al., 2020) and Distracting Control Suite (Stone et al., 2021) benchmark in generalization ans sample efficiency settings.

- Empirically, we observed that our agent ARPO performs better generalization and sample efficiency performance compared to standard PPO (Schulman et al., 2017), SAC (Haarnoja et al., 2018), and a data augmentation-based RAD (Laskin et al., 2020a) approach in various settings.

## 2 PRELIMINARIES AND PROBLEM SETTINGS

**Markov Decision Process (MDP)** An MDP is denoted by $\mathcal{M} = (\mathcal{S}, \mathcal{A}, \mathcal{P}, r)$ where at every timestep $t$, from an state $s_t \in \mathcal{S}$, the agent takes an action $a_t$ from a set of actions $\mathcal{A}$. The agent then receives a reward $r_t$ and the environment and move to a new state $s_{t+1} \in \mathcal{S}$ based on the transition probability $P(s_{t+1}|s_t, a_t)$.

**Reinforcement Learning**. Reinforcement learning aims to learn a policy $\pi \in \Pi$ that maps state to actions. The policy's objective is to maximize cumulative reward in an MDP, where $\Pi$ is the set of all possible policies. Thus, the policy which achieves the highest possible cumulative reward is the optimal policy $\pi^* \in \Pi$.

**Generalization in Reinforcement Learning**. In the zero-shot generalization framework (Song et al., 2020), we assume the existence of a distribution of levels $\mathcal{D}$ over an MDP and a fixed optimal policy $\pi^*$ which can achieve maximal return over levels generated from the distribution $\mathcal{D}$. The levels may differ in observational variety, such as different background colors. In this setup, the agent has access to a fixed set of MDP levels during training the policy. The trained agent is then tested on the unseen levels to measure the generalization performance of the agent.

**Style Transfer**. The task of image-to-image translation is to change a particular aspect of a given image to another, such as red background to green background. Generative adversarial network (GAN)-based method has achieved tremendous success in this task (Kim et al., 2017; Isola et al., 2017; Zhu et al., 2017; Choi et al., 2018). Given training data from two domains, these models learn to style-translate images from one domain to another. The domain is defined as a set of images sharing the same attribute value, such as similar background color and texture. The shared attributes in a domain are considered as the "style" of that domain. Many translation methods require paired images from two domains, which is not feasible in the reinforcement learning setup. The agent collects the data during policy learning, and annotating them in pairs is not feasible. Thus, we leverage unpaired image to image translation, which does not require a one-to-one mapping of annotated images from two domains.

In particular, we build upon the work of StarGAN (Choi et al., 2018) which is capable of learning mappings among multiple domains efficiently. The model takes in training data of multiple domains and learns the mappings between each pair of available domains using only a single generator. This method automatically captures special characteristics of one image domain and figures out how these characteristics could be translated into the other image collection, making it appealing in the reinforcement learning setup. However, the RL agent generates experience trajectory data which is

not separated into domains. Thus, we further apply a clustering approach which first clusters the trajectory observation into domains, and then we train the generator, which style-translates among those domains.

## 3 ADVERSARIAL ROBUST POLICY OPTIMIZATION (ARPO)

Our approach consists of a max-min game theoretic objective where a generator network modifies the observation by style transferring it to maximize the agent's probability of taking a different action. In contrast, the policy network (agent) updates its parameters to minimize the effect of perturbation while maximizing the expected future reward. Based on this objective, we propose a practical deep reinforcement learning algorithm, Adversarial Robust Policy Optimization (ARPO), to find a robust policy that generalizes to test environments. We now discuss the details of the policy network and generator.

### 3.1 POLICY NETWORK

The objective of the policy network is two-fold: maximize the cumulative RL return during training and be robust to the perturbation on the observation imposed by the generator. Ideally, the action probability of the translated image should not change from the original image, as the translated observation is assumed to represent the same semantic as the original observation.

The reinforcement learning loss $\mathcal{L}_\pi$ can be optimized with various reinforcement learning mechanisms, such as policy gradient (optimization). However, we do not make any assumption the type of RL algorithms to use for optimizing $\mathcal{L}_\pi$; thus, any algorithms can be incorporated with our method. Here we discuss the policy optimization formulation, which is based on the proximal policy optimization (PPO) (Schulman et al., 2017) objective:

$$\mathcal{L}_\pi = -\mathbb{E}_t\big[\frac{\pi_\theta(a_t|s_t)}{\pi_{\theta_{old}}(a_t|s_t)}A_t\big] \tag{1}$$

$$A_t = -V(s_t) + r_t + \gamma r_{r+1} + ... + \gamma^{T-t+1}r_{T-1} + \gamma^{T-t}V(s_T). \tag{2}$$

where, $\pi_\theta$ is the current policy and $\pi_{\theta_{old}}$ is the old policy; $A_t$ is estimated from the sampled trajectory by policy $\pi_{\theta_{old}}$ leveraging a value function estimator denoted as $V$. Here, both the policy $\pi$ and value network $V$ are represented as neural networks.

An additional objective of the policy is to participate in the adversarial game with the style transfer generator. Thus the final policy loss $\mathcal{L}_\theta$ is defined as follows:

$$\mathcal{L}_\theta = \mathcal{L}_\pi + \beta_1 KL[\pi_\theta(.|x_t), \pi_\theta(.|x'_t)] \tag{3}$$

The policy parameter $\theta$ is updated to minimize equation 3. For the observation $x_t$ at timestep $t$, the KL-component measure the distortion in the action distribution of the policy $\pi_\theta$ due to the perturbation ($x'_t$). The hyperparameter $\beta_1$ determines the participation in an adversarial objective with the generator. By minimizing the KL-part, the policy becomes robust to the change in perturbation proposed by the generator.

### 3.2 GENERATOR NETWORK

It takes the observation $x_t$ as input and outputs the style-translated observation $x'_t$. The objective of this network is to change the agent's (policy network's) probability of taking a different action for the corresponding input observation. However, the content or semantic of the observations needs to be intact, and the only style of the images will differ.

The first step of this process is to curate a dataset with different style features such as background color and texture of objects present in the observation. We first use experience trajectory observation data from the environment and separate them into different classes based on visual features (Figure 2). The task of the generator is then to translate the images from one class images to another class images. In this case, the "style" is defined as the commonality among images in a single class. We now discuss details about how we separate the trajectory observation to get different clusters.

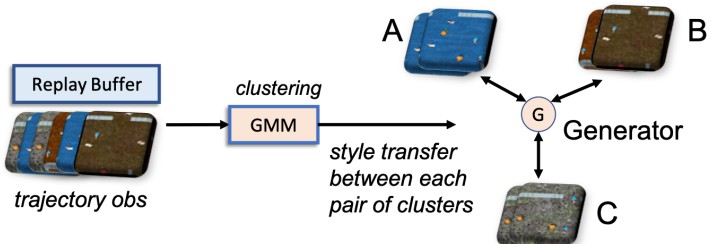

Figure 2: Overview of style transfer generator module. The experience trajectory observation data for the task environment are separated into different classes based on visual features using Gaussian Mixture Model (GMM) clustering algorithm. The task of the generator is then to translate the image from one class image to another classes images. In this case, the "style" is defined as the commonality among images in a single class. Given a new observation, it first infers into its (source) cluster using GMM, and then the generator translated it to (target) another cluster style. The target cluster is taken randomly from the rest of the cluster.

**Clustering Trajectory Observation**. The trajectory data is first clustered into $n$ clusters. Then, we use the Gaussian Mixture Model (GMM) for clustering and ResNet (He et al., 2016) for feature extraction and dimension reduction. Firstly, a pre-trained (on ImageNet) ResNet model reduces an RGB image into a 1000 dimensional vector. This step extracts useful information from the raw image and reduces the dimension, allowing faster cluster training. Note that this process focuses entirely on the visual aspect of the observation. After passing through the ResNet, the observation dataset is clustered using GMM.

**Generator Training**. Generator $G$ tries to fool discriminator $D$ in an adversarial setting by generating a realistic image represented by the true image distribution. We extend the generator's ability, and an additional objective of the generator is to fool the RL agent (policy network) by transforming the style of the original observation. For example, changing the background color to a different one does not change the semantic of the observation, but the image style would be different. Ideally, the action mapping of the policy needs to be the same on both the original and translated observation. However, a policy that focuses on the irrelevant background color will suffer from this translation, and thus its probability distribution deviates because of the style translation. This is defined as the KL-divergence $KL(\pi_\theta(.|x_t), \pi_\theta(.|x_t'))$, where $x_t$ is the given observation and $x_t'$ is the translated observation by the generator.

The generator tries to generate a style that eventually increases the KL quantity. On the other hand, the policy should learn to be robust toward such changes. The policy training described above addresses this and tries to minimize the KL-divergence due to the style transfer, which eventually allows robust policy learning and ultimately prevents the agent from overfitting irrelevant features from the high-dimensional observation.

We build on the generator on previous works (Choi et al., 2018; Zhu et al., 2017) which is a unified framework for a multi-domain image to image translation. The discriminator loss is

$$\mathcal{L}_D = -\mathcal{L}_{adv} + \lambda_{cls}\mathcal{L}_{cls}^r, \tag{4}$$

which consist of the adversarial loss $\mathcal{L}_{adv}$ and domain classification loss $\mathcal{L}_{cls}^r$. The discriminator detects a fake image generated by the generator G from the real image in the given class data.

On the other hand, the generator loss is

$$\mathcal{L}_G = \mathcal{L}_{adv} + \lambda_{cls}\mathcal{L}_{cls}^f + \lambda_{rec}\mathcal{L}_{rec} - \beta_2 KL(\pi_\theta(.|x_t), \pi_\theta(.|x_t')) \tag{5}$$

which consists of image translation components and policy component which is the KL-part. From the translation side, the $\mathcal{L}_{adv}$ is adversarial loss with discriminator $\mathcal{L}_{cls}^f$ is the loss of detecting fake image. Finally, a cycle consistency loss (Choi et al., 2018; Zhu et al., 2017) $\mathcal{L}_{rec}$ is used which makes sure the translated input can be translated back to the original input, thus only changing the domain related part and not the semantic. Some more details of these losses can be found in the supplementary materials.

Our addition to the generator objective is the KL-part in equation 5 which tries to challenge the RL agent and fool its policy by providing a more challenging translation. However, the eventual goal

of this adversarial game is to help the agent to be robust to any irrelevant features in the observations. The hyperparameter $\beta_2$ control the participation in the adversarial objective of the generator network.

Now we discuss a practical RL algorithm based on the above objectives.

### 3.3 ARPO Algorithm

The training of the policy network and the generator network happen in an alternating way. First, the policy is trained on a batch of replay buffer trajectory observation, and then the generator parameters are updated based on the objectives discussed above. The more overall training progresses, the better the generator at translating observation, and the policy also becomes better and becomes robust to irrelevant style changes. For stable training, the input images data for the generator is kept fixed initially, and we apply the GMM cluster only at the beginning to generate different class images (distribution). The pseudocode of the algorithm is given in Algorithm 1.

---

**Algorithm 1** Adversarial Robust Policy Optimization (ARPO)

---

1:  Initialize parameter vectors for policy network and generator network
2:  **for** each iteration **do**
3:      **for** each environment step **do**
4:          $a_t \sim \pi_\theta(a_t|x_t)$
5:          $x_{t+1} \sim P(x_{t+1}|x_t, a_t)$
6:          $r_t \sim R(x_t, a_t)$
7:          $\mathcal{D} \leftarrow \mathcal{D} \cup \{(x_t, a_t, r_t, x_{t+1})\}$
8:      **end for**
9:      **for** each observation $x_t$ in $\mathcal{D}$ **do**
10:         $x'_t \leftarrow Generator(x_t)$ // *generate translated observation*
11:         Compute $\mathcal{L}_\pi$ from data $\mathcal{D}$ using Equation 1, and 2 // *update policy for RL objective*
12:         Compute $\mathcal{L}_\theta$ using Equation 3 // *update policy with adversarial KL part*
13:         Update generator by computing Equation 6, and 5
14:     **end for**
15: **end for**

---

**Discussion on convergence**. The adversarial min-max KL component $KL(\pi_\theta(.|x_t), \pi_\theta(.|x'_t))$ become zero when the RL policy is optimal ($\pi^*$) and the generators only perturbs (translates) the irrelevant part of all observations. In that case, the optimal policy only focuses on the relevant part of the observations, that is, the true state; thus, any changes in irrelevant part due to style transfer will be ignored. At that point the KL component become zero as the $\pi_\theta^*(.|x_t) = \pi_\theta^*(.|x'_t)$. Note that in practice, the algorithm is usually trained for limited timesteps, and thus the adversarial training might not converge to optimal. However, empirically we observe that our algorithm ARPO achieves improved performance in many challenging Procgen and Distracting control suite tasks.

## 4 Experiments

### 4.1 Setup

Our experiments cover both discrete (Procgen) and continuous (Distracting control) action spaces with image-based observation.

**Procgen Environment**. We evaluate our proposed agent ARPO on the challenging RL generalization benchmark Procgen (Cobbe et al., 2020). We use the standard evaluation protocol from Cobbe et al. (2020) where the agent is allowed to train on only 200 levels of each environment, with the difficulty level set as *easy*. Then the agents are tested over more than 100K levels, full distribution. Thus, to better generalize to the test environment, the agent must master the skill and avoid overfitting to spurious features in the training environment. The observations are raw pixel images, and the environment varies drastically between levels; some snippets are given in Figure 3.

**Distracting Control Suite Environment**. The distracting control adds various visual distractions on top of the Deepmind Control suite (Tassa et al., 2020) to facilitate the evaluation of generalization

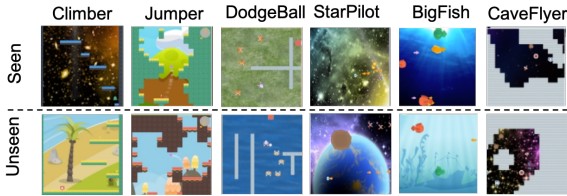

Figure 3: Some snippets of different Procgen environments (Cobbe et al., 2020). The training (seen) levels vary drastically from the testing (unseen) environment. The agent must master the skill without overfitting irrelevant non-generalizable aspects of the environment to perform better in unseen levels.

performance. We experimented on two distracting settings, *background (dynamic)* and *color* for *easy* difficulty mode. In the distracting background, videos played in the background, and for the color distraction, the body of the robot color changes in different agent interactions. The train and test environments differ by different distractions, such as separate videos for train and test. As the agent has to use image-based observation, the distractions make the task challenging. Some snippets of the environments can be found in Figure 4.

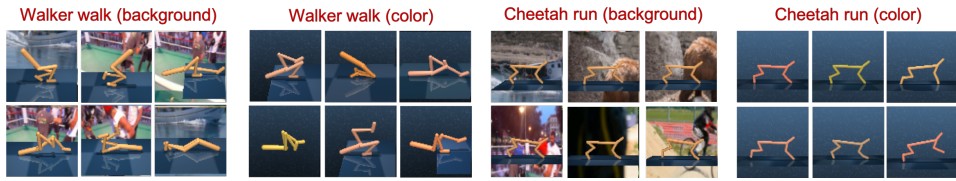

Figure 4: Some snippets from Distracting control suite (Stone et al., 2021).

**Baselines**. We compare our algorithm with Proximal Policy Optimization (**PPO**), an on-policy algorithm motivated by how we can quickly optimize our policy while staying close to the current policy. PPO performs effectively in many RL benchmarks, which includes the Procgen environments (Cobbe et al., 2020). Furthermore, we evaluated our method with a data augmentation technique, **RAD** (Laskin et al., 2020a) on Procgen environments. RAD is a plug-and-play module that can be used with existing RL algorithms and shows effective empirical performance in complex RL benchmarks, including some Procgen environments. In particular, we evaluated *Cutout Color* augmentation technique which has shown better results in many Procgen environments compared to other augmentation techniques evaluated in Laskin et al. (2020a). This augmentation is applied on top of standard PPO. Note that our ARPO algorithm can be easily plugged in with these data augmentation-based methods. However, in this paper, we evaluated how our ARPO agent performs even when no augmentation is used. Furthermore, in the Distracting control suite, in addition to PPO, we compare ARPO with Soft Actor-Critic (**SAC**) (Haarnoja et al., 2018) which is an off-policy algorithm that optimizes a stochastic policy.

**Implementation**. We implemented our ARPO on the Ray RLlib framework (Liang et al., 2018) and used a CNN-based model for the policy network for these experiments. For all the experiments with ARPO, we set the $\beta_1 = \beta_2 = 20$ (in equation 3, and 5). The generator's hyperparameters are set to $\lambda_{cls} = 1$, and $\lambda_{rec} = 10$. Number of cluster is set to 3. Details of model configurations and parameters are given in the Appendix. We will open-source our code and data for reproducibility and facilitating further research.

## 4.2 PROCGEN RESULTS

**Generalization**. The results in Figure 5 show the test results of ARPO and baselines PPO and RAD on 16 Procgen environments with image observation. Overall, ARPO performs better in many environments. We observe that, almost in half of the environments, our agent ARPO outperforms both baselines. It also performs comparably in a few other environments. Note that in some environments such as Ninja, Plunder, Chaser, Heist, all the agents fail to obtain a reasonable training accuracy (Cobbe et al., 2020) and thus perform poorly during testing (generalization).

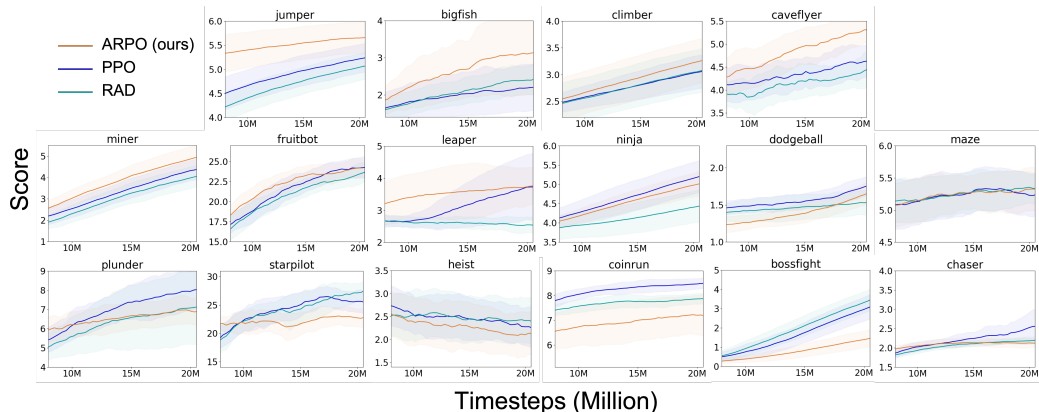

Figure 5: Generalization results. Test performance comparison on Procgen environments with image observation. ARPO performs better in many environments where it outperforms the baselines. The results are averaged over 3 seeds.

Table 1: Generalization results on Procgen environments. The results are the best test result achieved by each agent during the entire training timesteps. The mean value and standard deviation are calculated across 3 random seed runs. ARPO achieves the best generalization results in most of the environments compared to PPO and RAD. Best agent is in **Bold**.

| Env | ARPO (ours) | PPO | RAD |
|---|---|---|---|
| jumper | **6.5** $\pm$ 0.06 | 6.49 $\pm$ 0.15 | 6.35 $\pm$ 0.04 |
| bigfish | **4.18** $\pm$ 1.59 | 2.82 $\pm$ 0.75 | 3.22 $\pm$ 0.23 |
| climber | **5.88** $\pm$ 0.42 | 5.22 $\pm$ 0.59 | 5.53 $\pm$ 0.55 |
| caveflyer | **5.91** $\pm$ 0.08 | 5.5 $\pm$ 0.25 | 5.46 $\pm$ 0.34 |
| miner | **7.2** $\pm$ 0.71 | 6.8 $\pm$ 0.17 | 6.21 $\pm$ 0.36 |
| fruitbot | **27.04** $\pm$ 0.49 | 26.53 $\pm$ 0.57 | 26.99 $\pm$ 0.97 |
| leaper | 4.19 $\pm$ 0.55 | **4.33** $\pm$ 0.97 | 3.22 $\pm$ 0.06 |
| ninja | 6.49 $\pm$ 0.41 | **6.96** $\pm$ 0.5 | 5.82 $\pm$ 0.29 |
| dodgeball | **2.32** $\pm$ 0.22 | 2.28 $\pm$ 0.05 | 1.85 $\pm$ 0.17 |
| maze | 6.1 $\pm$ 0.14 | **6.23** $\pm$ 0.32 | 5.98 $\pm$ 0.04 |
| plunder | 8.37 $\pm$ 0.95 | **9.39** $\pm$ 0.47 | 8.34 $\pm$ 2.19 |
| starpilot | 29.0 $\pm$ 3.04 | 31.13 $\pm$ 0.27 | **31.59** $\pm$ 1.28 |
| heist | **3.93** $\pm$ 0.41 | 3.7 $\pm$ 0.35 | 3.75 $\pm$ 0.13 |
| coinrun | 7.67 $\pm$ 0.68 | **8.81** $\pm$ 0.12 | 8.25 $\pm$ 0.22 |
| bossfight | 4.08 $\pm$ 1.27 | 7.63 $\pm$ 0.6 | **7.99** $\pm$ 0.39 |
| chaser | 2.32 $\pm$ 0.09 | **3.11** $\pm$ 0.69 | 2.36 $\pm$ 0.04 |

This result shows that our proposed agent ARPO achieves a strong performance and reduces the generalization gap compared to the tested baselines on generalization.

We further show how each agent achieves generalization during the entire timesteps. Consequently, we compute the maximum score achieved by each agent in the entire duration and take the average best score. Finally we report mean and standard deviation of these best scores across 3 seed runs. The generalization results are computed by evaluating the trained agents on test levels (full distribution). The performance is computed by evaluating each agent for 10 random episode trials after a training interval of 10. All agents ran for 22 million timesteps for training. Table 1 shows the results comparison for all the agents on Procgen environments. Similar to the previous discussion, we see that in many of the environments, our ARPO agents perform better than the baselines. ARPO performs better than both the baselines in half of the environment, and in 12 out 16 environments, ARPO performs better than at least on baselines. This result shows that our agent outperforms the baselines in the generalization results on Procgen benchmarks.

**Sample Efficiency**. We evaluate and compare our agent on the sample efficiency during training. This result shows how quickly our agent achieves a better score during training. Learning curve can be found in Appendix B (Figure 7).

In many environments, we see our agent ARPO perform better than the baselines PPO and RAD. These results show that despite optimizing for the generalization, our adversarial training helps the ARPO agent learn a robust policy which is also better during training.

### 4.3 DISTRACTING CONTROL RESULTS

Figure 6 shows the results of ARPO and PPO on Walker walk environment from distracting control suite. We observe that ARPO performs better compared to the baseline algorithm PPO in both

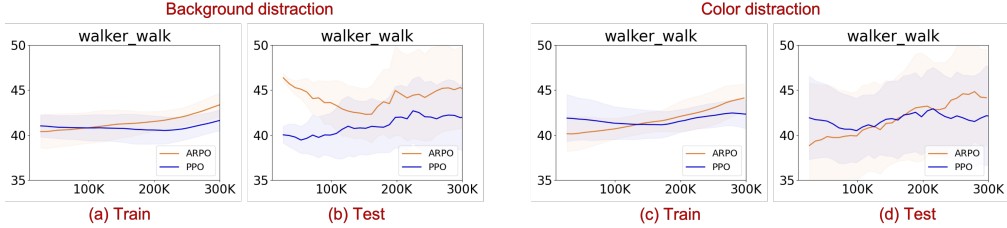

Figure 6: Sample efficiency (Train) and generalization (Test) results on Distracting control Walker walk environments. The results are averaged over 3 seeds.

sample efficiency (train) and generalization (test) for background and color distraction. Note that we observe large variance (standard deviation) in the results across the run, particularly in test time. These results correspond to the benchmark results where Stone et al. (2021) also find comparatively larger variance in reported results. In some cases (Figure 6b), the test performance surpasses the corresponding timestep's train reward. This scenario might happen because the videos for train and test environments have different difficulties. Thus in some cases, the agent finds better test performance than the training. However, our ARPO's performance remains better during testing compared to PPO. We also observe that in these settings ARPO (and also PPO) performs better compared to SAC. The results are in the Appendix D (Figure 10). Furthermore, results analysis for Cheetah run environment is in the Appendix C.

Please see the Appendix for ablation study (Section E), and related work (Section F).

## 5 DISCUSSION

Lack of visual diversity in the observation of the environment might result in poor clustering, which eventually leads to less challenging style translation by the adversarial generator. However, in those scenarios, the reinforcement learning algorithms often perform well as the train and test environment remain consistent (e.g., same background). Furthermore, this setup might lead the agent to overfit the training environment, which results in the agent performing poorly in a slightly different environment (Song et al., 2020; Zhang et al., 2018b). In this paper, we are interested in the scenario where the train and test environment are visually different, which is a more practical setup (Cobbe et al., 2020; Stone et al., 2021).

In summary, we proposed an algorithm, Adversarial Robust Policy Optimization (ARPO), to improve generalization for reinforcement learning agents by removing the effect of overfitting in high-dimensional observation space. Our method consists of a max-min game theoretic objective where a generator is used to transfer the style of high-dimensional observation (e.g., image), thus perturb original observation while maximizing the agent's probability of taking a different action for the corresponding input observation. In contrast, a policy network updates its parameters to minimize the effect of such translation, thus being robust to the perturbation while maximizing the expected future reward. We tested our approaches on Procgen and distracting control benchmarks in generalization setup and confirmed the effectiveness of our proposed algorithm. Furthermore, empirical results show that our method generalizes better in unseen test environments than the tested baseline algorithms.

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

# A  APPENDIX

# B  ADDITIONAL PROCGEN RESULTS

Figure 7 shows the sample efficiency results for ARPO, PPO, and RAD.

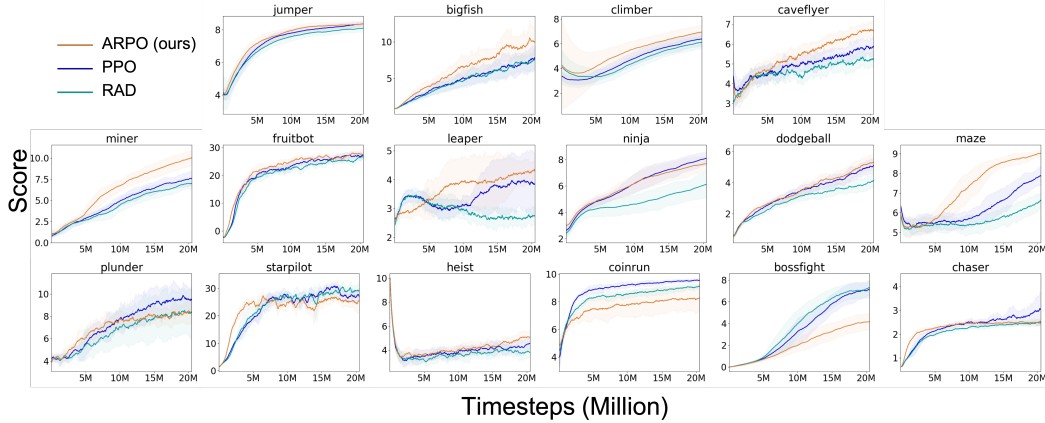

Figure 7: Sample efficiency results. Train performance (learning curve) comparison on Procgen environments. Despite using perturbation on the observation, our agent ARPO outperforms the baselines in many environments, thus achieving better sample efficiency during training. The results are averaged over 3 seeds.

**Procgen aggregated results**. We further show the aggregated results on all the environments in the Procgen environment. Figure 8 shows the aggregated results comparison on all 16 environments of Procgen, which is a probabilistic measure that highlights how likely an improvement of the algorithm over other proposed in Agarwal et al. (2021b).

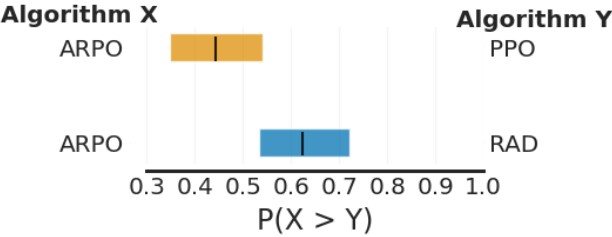

Figure 8: Probabilistic measure (Agarwal et al., 2021b) of how likely our agent ARPO is to improve over the baseline PPO and RAD. We observe that ARPO is up to 54% likely to improve over PPO and up to 72% likely to improve over baseline RAD in the Procgen generalization setup.

## C   ADDITIONAL DISTRACTING CONTROL RESULTS

Results for ARPO and PPO on Cheetah run with background distraction are in Figure 9. We see that both agents perform comparably in both background and color distraction. We observe similar results for both the agents in this setup of Cheetah run environments. Note that in this training timestep the PPO agents perform poorly in the training and testing. The highest achievable reward for this environment is much higher (see Stone et al. (2021)). Consequently the the ARPO agent could not improve over the base algorithm PPO. This suggest that the base algorithm needs to trained sufficiently to achieve any reasonable reward before we benefit of the adversarial training.

Note that the resulting score is low compared to the reported results in the benchmark paper Stone et al. (2021) which suggests the PPO itself could not learn helpful behavior in this environment.

## D   COMPARISON WITH SAC

Results comparison on Distracting Control for ARPO, PPO, and SAC are in Figure 10 (Walker walk), and 11 (Cheetah run). We see that ARPO (and PPO) perform better compared to the SAC in both sample efficiency (train) and generalization (test) in these settings.

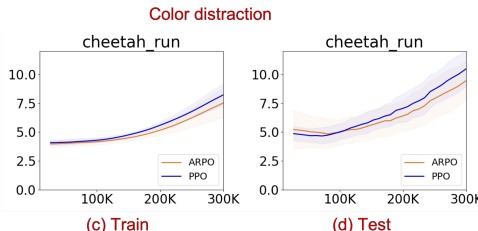

Figure 9: Distracting control Cheetah run environment results. The results are averaged over 3 seeds. **(a, c)** Sample efficiency (train), and **(b, d)** Generalization (test).

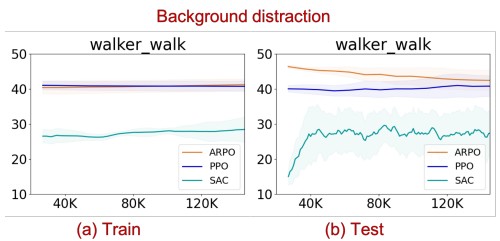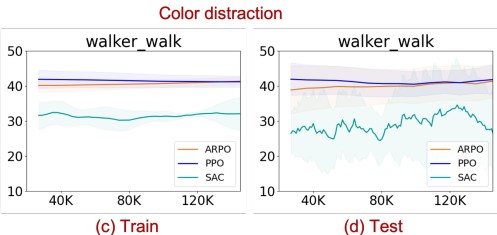

Figure 10: Distracting control Walker walk environment results (including SAC). The results are averaged over 3 seeds. **(a, c)** Sample efficiency (train) **(b, d)** Generalization (test).

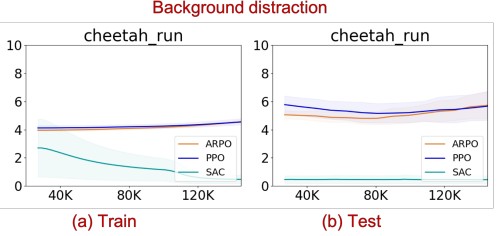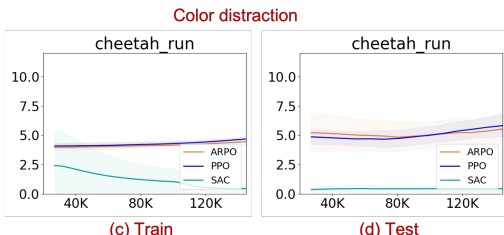

Figure 11: Distracting control Cheetah run environment results (including SAC). The results are averaged over 3 seeds. **(a, c)** Sample efficiency (train), and **(b, d)** Generalization (test).

# E  ABLATION STUDY

## E.1  HYPERPARAMETERS OF ARPO

We conduct a study on how ARPO agent's performance varies due to its hyperparameters: (a) the number of clusters generated by the GMM model for generator training, and (b) $\beta_1$, $\beta_2$ which indicate the amount of participation of policy and generator in the adversarial objective.

Figure 12 shows the ablation results. We observe that our ARPO agent shows improvement in generalization (test) performance with the increase in number of cluster on Climber environment (Figure 12a). As the number of cluster increase, the generator has more option to choose for translation, and thus the policy might face a hard challenge and thus learn a more robust policy. In this paper, we report the ARPO agent's results on cluster $n\_cluster = 3$. On the other hand, and ablation results on different $\beta$ values are shown in Figure 12b. When $\beta$ values are large the test performance goes up; however, training performance suffer a bit. In this paper, we report the results on $\beta_1 = \beta_2 = 20$, which shows a balance in train and test performance.

## E.2  QUALITATIVE ANALYSIS OF ADVERSARIAL STYLE TRANSFER

In this section we discuss how the generator performs in generating adversarially translated observation during agent training.

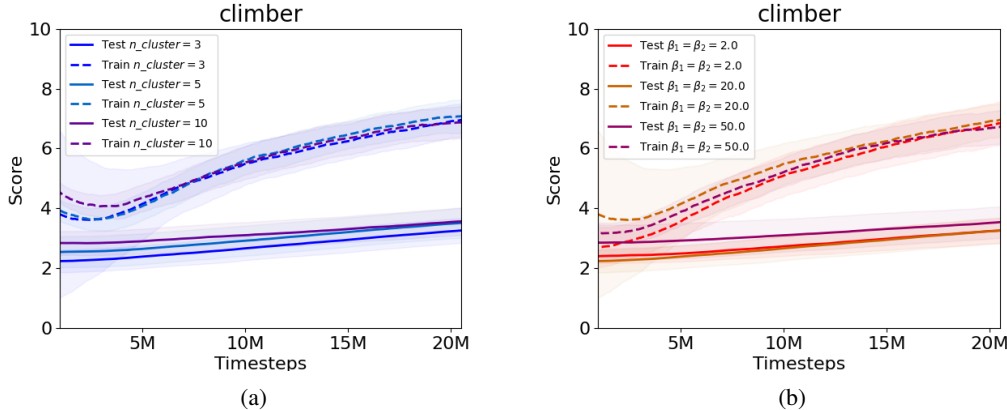

(a)                            (b)

Figure 12: Ablation results. The results are averaged over 3 seeds. **(a)** Our ARPO agent's results on different cluster numbers. It improves generalization (test) performance with the increase in cluster number in the Climber environment. **(b)** Our ARPO agent's results on different $\beta$ values which determine participation on adversarial optimization.

Figure 13, 14, 13, 15, and 16 show sample translation from the generator. Please see the corresponding caption for detailed discussion.

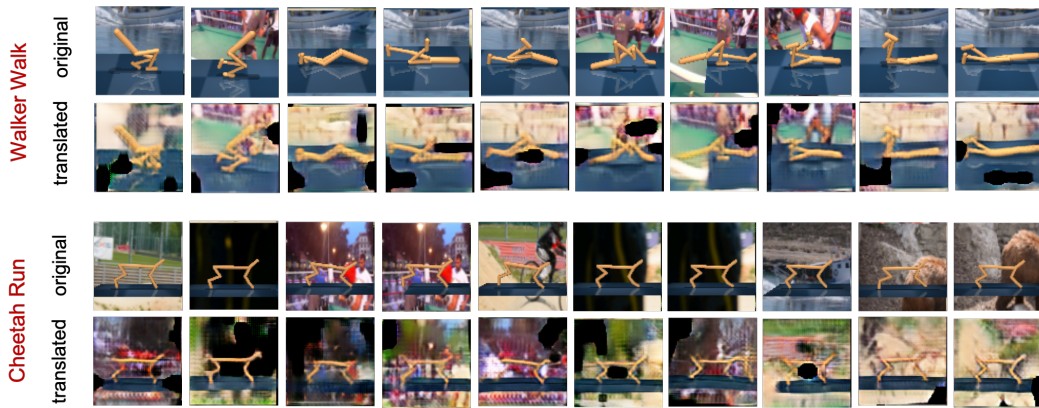

Figure 13: Sample translation of generator on **background** distraction for Walker-walk environment from Distracted Control Suite benchmark. We see that the translated observations retain the semantic that is the robot pose while changing non-essential parts. Interestingly, we observe that the adversarial generator blackout some non-essential parts of the translated images. This scenario might indicate that the min-max objective between policy network and generator tries to recover the actual state from the observation by removing parts irrelevant to the reward.

### E.3 KL REGULARIZATION

Figure 17 shows the KL regularization results in different policy iteration. We for each environment we run the experiment and collect KL value ($KL(\pi_\theta(.|x_t), \pi_\theta(.|x_t'))$) for each policy iteration. To calculate moving average, we select a window size of 500 and the standard deviation is calculated on these 500 values, and showed in the shaded area in Figure 17.

## F RELATED WORK

**Data augmentation in RL**. In recent time, many approaches have been proposed to address generalization challenges including various data augmentation approaches (Cobbe et al., 2019; Laskin et al., 2020a; Kostrikov et al., 2020; Raileanu et al., 2020; Laskin et al., 2020b; Igl et al., 2019;

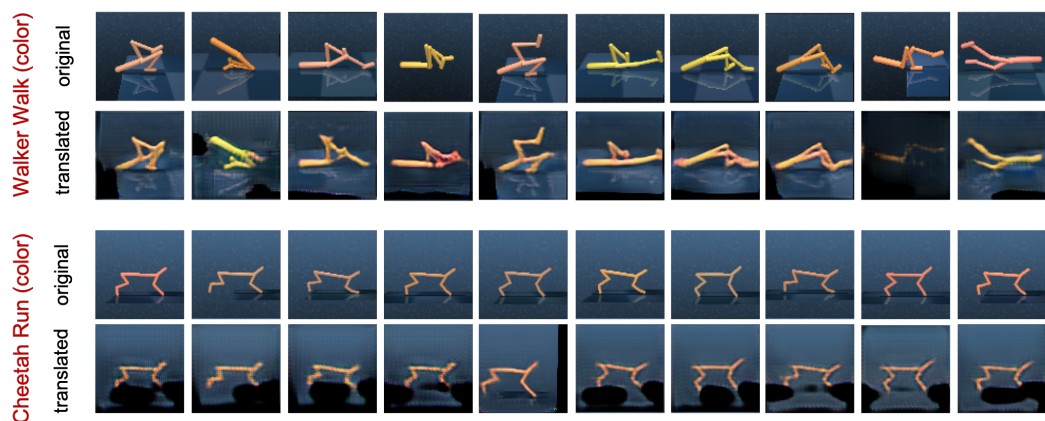

Figure 14: Sample translation of generator on **color** distraction for Walker-walk environment from Distracted Control Suite benchmark.

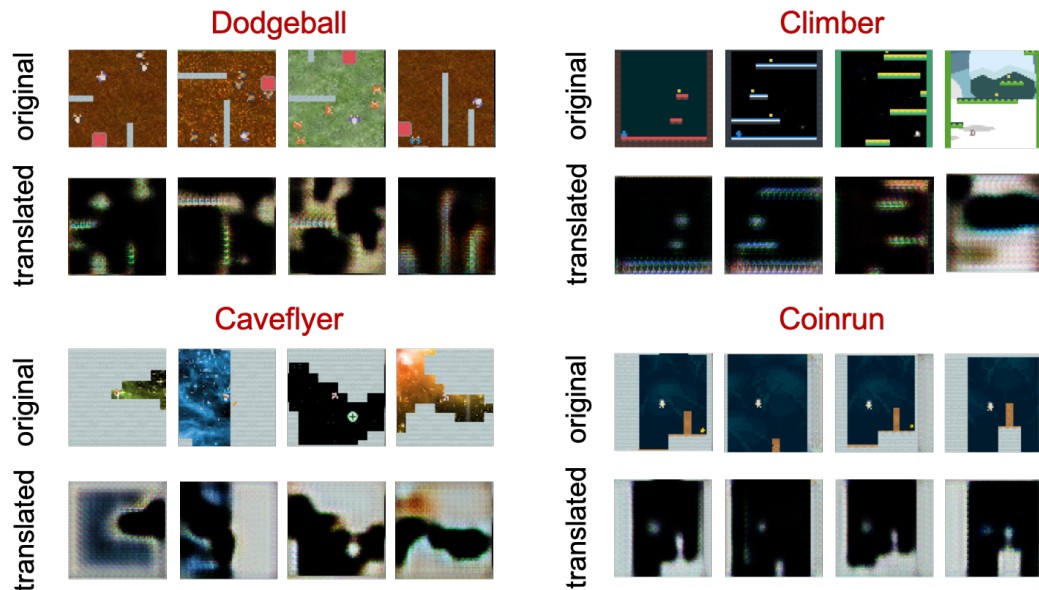

Figure 15: Sample translation of generator on four Procgen environments. We see that the generator retains most of the game semantic while changing the background color and the texture of various essential objects.

Osband et al., 2018; Lee et al., 2020; Wang et al., 2020; Zhang et al., 2021) have shown to improve generalization. The common theme of these approaches is to increase diversity in the training data; however, these perturbations are primarily performed in isolation of the RL reward function, which might change crucial aspects of the observation, resulting in sub-optimal policy learning. In contrast, we propose a style transfer that tries to generate semantically similar but different visual style observation to train a robust policy. Furthermore, our style perturbation takes into account the reward signal from the policy.

Note that the base RL algorithm (e.g., PPO) in our ARPO method still uses the original observation from the environment while participating in adversarial objectives (see Figure 1). Thus, data augmentation can be applied to the original observations before feeding them to the base RL algorithm to potentially improve the base RL agent, as observed in many data augmentation techniques discussed above.

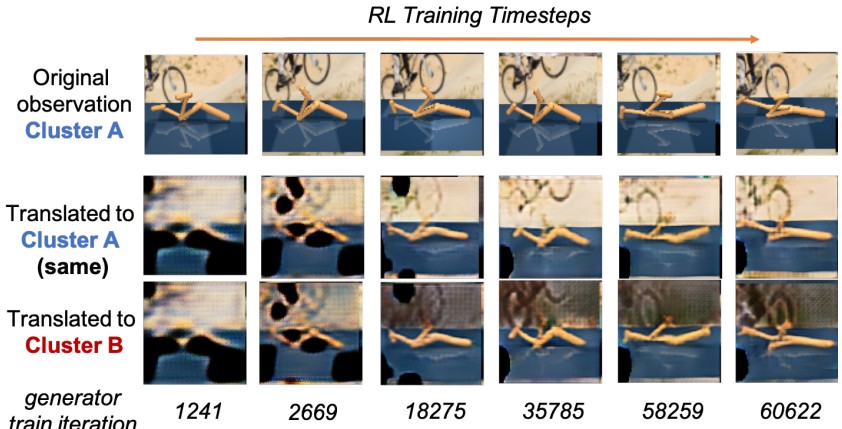

Figure 16: Sample translation of generator during various training phases on **background** distraction for Walker-walk environment from Distracted Control Suite benchmark. [**Second row**] We see that when translated back to the same cluster (in the second row), the generated images get some distortion (e.g., blackout some parts), while the essential parts remain mostly intact. This scenario might be happening due to the regularization of the cycle consistency loss by the KL component in equation 5. [**Third row**] We see that the translation to a different cluster generates images that changes the irrelevant parts while mostly keeping the semantic intact. In both the second and third row, the translation gets better as time progresses, suggesting the adaptation of both policy and generator networks due to the min-max objective.

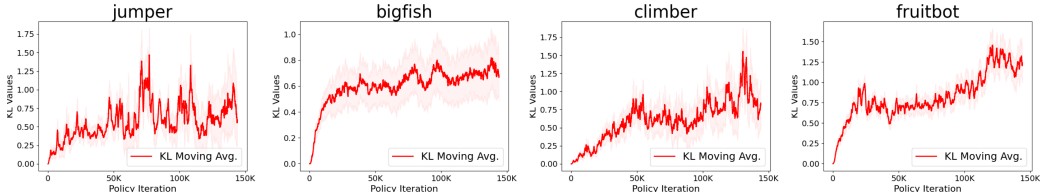

Figure 17: KL regularization during RL training for four Procgen environments. At the beginning of the training, both the policy and generator behave randomly; thus, the KL values become nearly zero. Gradually, the KL started to increase as the policy learning progressed with the adversarial generated observation. Eventually, the policy tries to adjust to the changes in the observation, and the KL values become stable (**bigfish**) or going downward (**jumper**, and **climber**). We see that in **fruitbot** environment, the values started to increase at the end, which suggests a possible divergence. This scenario might be an explanation of why the generalization performance started to drop at the end in Figure 5, possible overfitting. Thus this KL measure might be a tool to detect possible divergence and overfitting. However, each environment's stability cutoff time (policy iteration) is different, suggesting that the KL regularization behavior might be different depending on the environment complexity. Ideally, with enough training, the KL values should converge to zero (again) if the policy reaches optimal (for true MDP state) and the generator fully recovers the true state from the observation and changes all the irrelevant information from the observation. Note that we limit the policy to train until a cutoff time (22 Million RL timesteps).

**Style Transfer in RL**. Many approaches have been proposed which use style transfer to improve reinforcement learning algorithms. Gamrian & Goldberg (2019) use image-to-image translation between domains to learn the policy in zero-shot learning and imitation learning setup. They use different levels' data and train unpaired GANs between two levels (domains), which requires access to the annotated agent's trajectory data in both source and target domains. Furthermore, Smith et al. (2019) performs pixel-level image translation via CycleGAN to convert the human demonstration to a video of a robot. These translated data can then be used to convert the human demonstration into a video of a robot used to construct a reward function that helps learn complex tasks. However, their CycleGAN operates on two predefined domains, human demonstration and robot video.

In contrast, we first automatically generate the domains using a clustering algorithm and then train the style-translation generator. In our case, we do not need the information of levels; instead, we automatically cluster the trajectory data based on visual features of the observations. Additionally,

our visual-based clustering may put observations from multiple levels into a cluster, potentially preventing GAN from overfitting Gamrian & Goldberg (2019) to any particular environment levels.

**Adversarial Approach in RL**. Adversarial methods have been used in the context of reinforcement learning to improve policy training. Li et al. (2021) proposes to combine data augmentation with auxiliary adversarial loss to improve generalization. In particular, they use an adversarial discriminator to predict the label of the observation. However, they need to know the domain label, limiting the applicability as it might not be available, and the number of labels might be large, for example, in procedural generation. In contrast, we use an unsupervised clustering that automatically finds out similar visual features later used for generator training. (Pinto et al., 2017b) use adversarial min-max training to learn robust policy against forces/disturbances in the system (Pinto et al., 2017b). In addition, adversarial training has been used between two robots to improve object grasping (Pinto et al., 2017a) and in the context of multi-agent self-play scenario (Heinrich & Silver, 2016). Zhang & Guo (2021) generate adversarial examples for observations in a trajectory by minimizing the cumulative reward objective of the reinforcement learning agent. In contrast to these methods, we use adversarial visual-based style-transfer of the observation to guide robust policy learning by enforcing the policy to produce similar output action distributions.

**Generalization in RL**. Many methods have been proposed to reduce the generalization gap between training and unseen test environment in the context of reinforcement learning. State learning approaches (Higgins et al., 2017; Agarwal et al., 2021a; Zhang et al., 2020) and auto-encoder based latent state learning (Lange et al., 2012; Lange & Riedmiller, 2010; Hafner et al., 2019) with reconstruction loss have been proposed. The sequential structure in RL (Agarwal et al., 2021a) and behavioral similarity between states (Zhang et al., 2020) to learn invariant states have been leveraged to improve RL robustness and generalization. In contrast, we learn the invariant state by providing the RL agent with an additional variant of the same observation while retaining the reward structure using an adversarial max-min objective.

Our primary focus is to demonstrate how the adversarial style transfer helps learn a robust policy and improves the robustness of the base RL policy used. Thus our method is orthogonal to many existing algorithms, including strong baseline IDAAC (Raileanu & Fergus, 2021). Combining our method with these algorithms could improve performance; investigating them can be an interesting avenue for future work.

## G   TRAINING LOSS OF STARGAN GENERATOR

The adversarial loss is calculated as the Wasserstein GAN (Arjovsky et al., 2017) objective with gradient penalty which is defined as

$$\mathcal{L}_{adv} = \mathbb{E}_x[D_{src}] - \mathbb{E}_{x,c}[D_{src}(G(x,c))] - \lambda_{gp}\mathbb{E}_{\hat{x}}[(||\nabla_{\hat{x}}D_{src}(\hat{x})|| - 1)^2], \tag{6}$$

where $\hat{x}$ is sampled uniformly along a straight line between a pair of real and generated fake images. Here, the gradient penalty helps to stabilize the GAN training process and has been shown to perform better than traditional GAN training (Choi et al., 2018; Arjovsky et al., 2017; Gulrajani et al., 2017). We set the hyperparameter $\lambda_{gp} = 10$.

The classification loss of real image is defined as

$$\mathcal{L}_{cls}^r = \mathbb{E}_{x,c'}[-\log D_{cls}(c'|x)], \tag{7}$$

where $D_{cls}(c'|x)$ is the probability distribution over all clusteer labels. Similarly, the classification loss of fake generated image is defined as

$$\mathcal{L}_{cls}^f = \mathbb{E}_{x,c}[-\log D_{cls}(c|G(x,c))], \tag{8}$$

The reconstruction loss with the generator objective defined as

$$\mathcal{L}_{rec} = \mathbb{E}_{x,c,c'}[||x - G(G(x,c),c')||_1], \tag{9}$$

where we use the $L1$ norm.

Table 2: Hyperparameters for Procgen (RLlib) Experiments

| Description | Hyperparameters |
|---|---|
| Discount factor of the MDP | $gamma : 0.999$ |
| The GAE(lambda) parameter | $lambda : 0.95$ |
| The default learning rate | $lr : 5.0e - 4$ |
| Number of epochs per train batch | $num\_sgd\_iter : 3$ |
| Total SGD batch size | $sgd\_minibatch\_size : 2048$ |
| Training batch size | $train\_batch\_size : 16384$ |
| Initial coefficient for KL divergence | $kl\_coeff : 0.0$ |
| Target value for KL divergence | $kl\_target : 0.01$ |
| Coefficient of the value function loss | $vf\_loss\_coeff : 0.5$ |
| Coefficient of the entropy regularizer | $entropy\_coeff : 0.01$ |
| PPO clip parameter | $clip\_param : 0.2$ |
| Clip param for the value function | $vf\_clip\_param : 0.2$ |
| Clip the global amount | $grad\_clip : 0.5$ |
| Default preprocessors | $deepmind$ |
| PyTorch Framework | $framework : torch$ |
| Settings for Model | $custom\_model : impala\_cnn\_torch$ |
| Rollout Fragment | $rollout\_fragment\_length : 256$ |

## H    ENVIRONMENT DETAILS

**Procgen** We conducted experiments on OpenAI Procgen Cobbe et al. (2020) benchmark, consisting of diverse procedurally-generated environments with different action sets. This environment has been used to measure how quickly (sample efficiency) a reinforcement learning agent learns generalizable skills. These environments greatly benefit from the use of procedural content generation, the algorithmic creation of a near-infinite supply of highly randomized content. The design principles consist of high diversity, tunable difficulty, shared action, shared observation space, and tunable dependence on exploration. Procedural generation logic directs the level layout and other game-specific details. Thus, to master any of these environments, agents must learn an effective policy across all environment variations. We use all 16 environments available in this benchmarks.

All environments use a discrete 15 dimensional action space which generates $64 \times 64 \times 3$ RGB image observations. Note that some environments may use no-op actions to accommodate a smaller subset of actions.

## I    IMPLEMENTATION DETAILS

**Procgen Experiments** For experimenting on Procgen environments, we used RLlib Liang et al. (2018) to implement our ARPO, and baselines PPO and RAD cutour color algorithms. For all the agents' policy network (model), we use a CNN architecture used in IMPALA Espeholt et al. (2018) which is the best performing model in Procgen benchmark (Cobbe et al., 2020). We use the same policy parameters for all agents for a fair comparison.

Policy learning hyperparameter settings for all the agents (ARPO, PPO, and RAD) are set same for fair comparison and detect the effect of our proposed method (max-min adversarial objective with perturbation network). These hyperparameters are given in Table 2. Note that only the custom parameters are given here, other defaults parameter values can be found in the RLlib library Liang et al. (2018).

**Distracting Control Experiments** For experimenting on Distracting Control Suite (Stone et al., 2021) environments, we used RLlib Liang et al. (2018) to implement our ARPO, and baselines PPO and SAC algorithms. We use the PPO's CNN-based policy network (model) from RLlib for ARPO, and PPO. The policy specific parameters for ARPO, and PPO are the same which are given in Table 3.

For SAC, we use it's CNN-based model available in RLlib. The policy specific parameters are given in Table 4. Note that only the custom parameters for the RLlib implementation are given here, other defaults parameter values can be found in the RLlib library Liang et al. (2018).

Table 3: ARPO, and PPO Hyperparameters for Distracting Control Experiments

| Description | Hyperparameters |
|---|---|
| Number of epochs per train batch | $num\_sgd\_iter : 3$ |
| Total SGD batch size | $sgd\_minibatch\_size : 256$ |
| Training batch size | $train\_batch\_size : 8192$ |
| PyTorch Framework | $framework : torch$ |

Table 4: SAC Hyperparameters for Distracting Control Experiments

| Description | Hyperparameters |
|---|---|
| Training batch size | $train\_batch\_size : 512$ |
| Timesteps per iteration | $timesteps\_per\_iteration : 1000$ |
| Timesteps per iteration | $learning\_starts : 5000$ |
| PyTorch Framework | $framework : torch$ |

**Computing details**. We used the following machine configuration to run our experiments: 20 core-CPU with 256 GB of RAM, CPU Model Name: Intel(R) Xeon(R) Silver 4114 CPU @ 2.20GHz, and a Nvidia A100 GPU.

