# OpenReview forum: "Adversarial Style Transfer for Robust Policy Optimization in Reinforcement Learning"
_ICLR.cc/2022/Conference — ICLR 2022 Submitted_

### Official Review · Reviewer_FV5x · 2021-10-27

**Correctness:** 3
**Technical Novelty And Significance:** 3
**Empirical Novelty And Significance:** 2
**Recommendation:** 5
**Confidence:** 5

**Main Review:**

# Strengths
- The idea of using style transfer combined with adversarial perturbation of the visual input to improve generalisation in RL is interesting and novel.
- The method is benchmarked in a standard way on a standard benchmark, which aids reproducibility and comparison.
- The description of the approach is fairly clear, and if code is released then I believe the method would be quite reproducible

# Weaknesses
- The paper oversells it's empirical results. In the abstract it claims to outperform SOTA algorithms, but SOTA on Procgen is not PPO or RAD, but IDAAC (https://arxiv.org/abs/2102.10330). Further, RAD isn't even the SOTA method which applies data augmentation on Procgen, as that's UCB-DrAC (https://arxiv.org/abs/2006.12862). In many places the authors claim the method consistently outperforms the baselines, but they only perform better than both the baselines in 8 out of 16 games in terms of test performance, and many of those improvements aren't outside the confidence intervals. I think the language around the comparisons to baselines should be made less grand and more accurate.
- Further, it would be beneficial to actually compare to SOTA results, which can be taken from the IDAAC paper https://arxiv.org/abs/2102.10330. Or justify why your method shouldn't be compared against SOTA results
- The presentation of results in Table 1, where their method is bolded if it outperforms a single baseline, is non-standard and unfairly promotes their method (the other methods aren't bolded if they outperform any other method). I think this practice shouldn't be done.
- The method implicitly relies on the visual inputs being able to be clustered into distinct visual themes, which Procgen does have, but many other methods do not. It would be worth investigating whether this method works on benchmarks with less distinct clusters of visual themes.
- In general, no limitations of the method are discussed, and I think they should be to ensure that the method's strengths and weaknesses are fully understood.
- The related work section could use more discussion of other approach which take an adversarial approach to improving generalisation, such as https://arxiv.org/abs/2102.07097, https://arxiv.org/abs/2106.15587 and https://arxiv.org/abs/1703.02702.

# Questions
- In table 1, what do you mean by "the mean of the best test results"?

# Suggestions for improvements.

## Large:
- Evaluating the method on another generalisation benchmark, such as distracting control suite (https://arxiv.org/abs/2101.02722), would better validate the usefulness of this method. I think the idea of using style-transfer for adversarial data augmentation is interesting, and so I don't think that SOTA results are required, but showing improved results on a range of benchmarks would add evidence for the method being generally applicable and useful.
- In particular, the method seems useful from two perspectives: (1) from a sample-efficiency point of view as data augmentation and auxiliary representation learning losses often increases that. (2) from a generalisation perspective, *if* the training environment doesn't contain every combination of visual style and underlying state-space level (as is the case for procgen). I think an evaluation protocol for Distracting Control Suite could disentangle these two perspectives further.

## Small:
- Typo: Pg 5 "irreverent" -> "irrelevant"
- Reporting min-max normalised score, aggregated across games, and displaying performance profiles, as motivated in https://arxiv.org/abs/2108.13264, would improve the results reporting dramatically, and allow a scalar comparison between their method and the baselines.
- When using it as a verb I think you should write "style-translate", or just "translate the style of..." rather than "style translate" without the hyphen.
- Describing Procgen as "Visually enriched" implies you've changed (enriched) the procgen environment with more visual variety, which I don't think you have. Procgen is an established benchmark, I don't think you need to prefix it with adjectives when you mention it. Further, you describe the evaluation protocol you use as if you created it, but it's the standard evaluation protocol for easy mode in Procgen, which I think is worth mentioning.
- In the related work section, it's unclear why Burda et al 2018 is in the list of approaches using data augmentation (as it targets exploration, not generalisation), and Cobbe et al 2019 is in the list twice, and Kostrikov et al 2020 three times.

**Summary Of The Paper:**

The paper propose a method for improving generalisation and robustness across visual factors of variation in reinforcement learning, using a style transfer network to adversarially perturb the input to the policy, while the policy is trained to be invariant to this visual perturbation. The style transfer network is trained using StarGAN, based on domains created through a Gaussian Mixture Model clustering of the observations from the policy's experience. The method is demonstrated on OpenAI Procgen easy mode, and compared against PPO and RAD. The method does better than both of these baselines on 8 of the 16 games. Ablations on a single game show that the method is fairly robust to different choices of hyperparameters (number of clusters for the GMM, and weight of the adversarial loss for the style transfer network and policy).

**Summary Of The Review:**

While I think the idea motivating this paper is interesting and potentially useful, I think the execution of the idea in this paper, the lack of strong or varied empirical results, the overselling of the effectiveness of the method, and the incomplete discussion of related works makes me recommend the paper is rejected. Several of these criticisms are relatively easy to address, but overall I'm unlikely to recommend acceptance unless the authors add encouraging empirical results on a different benchmark, or vastly improve the results they currently have.

EDIT: after seeing the response from the authors I have raised my score from 3 to 5, and my correctness score from 2 to 3. I believe the improvements are worthwhile but don't make the paper worthy of acceptance: crucially, I think better comparisons with baselines (such as UCB-DrAC, and a more performant SAC implementation), and combining the method with an off-policy method (such as SAC or QT-Opt) would make the empirical justification of the idea more robust and make me more likely to accept the paper.

---

> ### Author Response · Authors · 2021-11-21
> **Response to Reviewer FV5x - Part 1**
>
> Thank you for the insightful comments. We have addressed your comments here and updated the paper accordingly. Please let us know if you have any further concerns about this paper.
>
> **Paper claims and SOTA** \
> We have modified the claim in the abstract (see end of the abstract), introduction (see end of the introduction section), and results discussion (Section 4.2). We make our statement more specific. Please refer to the updated paper.
>
> Our primary focus is to demonstrate how the adversarial style transfer helps learn a robust policy and improves the robustness of the base RL policy used. Thus our method is orthogonal to many existing algorithms, including strong baseline IDAAC. Combining our method with these algorithms could improve performance; investigating them can be an interesting avenue for future work.
>
> **Presentation of results in Table 1:** \
> We have modified the Table 1 and bold the best agents for all the environments. Thanks for the suggestion.
>
> **Limited visual diversity** \
> We have added the results of color distraction from the distracting control suite. In this setting, the visual diversity is less (as only the color of the robot body varies across episodes) than Procgen and background distraction in distracting control suite. On the walker walk environment, we observe an improvement of ARPO over baseline PPO.
> However, in the setups where visual diversity is limited, our method might face challenges. We have added a discussion on this limiting factor in the updated paper's Discussion section (previous Conclusion).
>
> **Discussion on limitation** \
> Thanks for the suggestion. We have added a discussion on the limitation in the updated paper's Discussion section (previous Conclusion).
>
> **Related work:** \
> We have revised the related work section based on your suggestion. Please refer to the related work section (now in the Appendix Section F, because of the space limitation) for a detailed discussion. In addition, we have added discussion on your suggested papers except for https://arxiv.org/abs/2106.15587, for which we could not find a peer-reviewed version. Please let us know if you have a reference for a peer-reviewed version of this paper. We will be happy to add a discussion on it in the paper.
>
> **Mean of the best test results:** \
> It is the best test score of the agent achieved during the entire timesteps. We have modified the Table 1 caption to reflect this.
>
> **Evaluating the method on another generalization benchmark:** \
> Thank you for your suggestion. We have included the experiments on the distracting control suite in the experiment section (Setup in Section 4.1, results in Section 4.3). Figure 6 shows that our method ARPO performs better compared to baseline PPO on Walker walk environments in both sample efficiency (train) and generalization (test) on two variations of distraction (background and color). Additional results are in Appendix C. We also show qualitative analysis of the generator's style translation in Appendix E.2. With these new results, we now show the improved results of our method on two benchmarks with both discrete and continuous actions spaces with image-based observation. These results demonstrate the usefulness and general applicability of our method.
>
> **Evaluation protocol for Distracting Control Suite:** \
> Thank you for suggesting distracting control environment which is well aligned with our methods. We add new results and analysis on this benchmark. Overall, our method performs better in the Walker walk environment in various settings compared to the baseline RL algorithm PPO. Please refer to the updated paper for more discussion on these results.
>
> We think the additional results and discussion can help in understanding the effectiveness of our proposed method.
>
> **Typo:** \
> We have fixed the typo, thank you.
>
> **Normalized Score:** \
> Thank you for your suggestion. We have added a normalized probabilistic measure from the suggested paper[1] on the Procgen generalization results in Appendix B, Figure 8. These are aggregated results overall Procgen environments and it measures how likely our agent ARPO is to improve over the baseline PPO and RAD. We observe that ARPO is up to 54\% likely to improve over PPO and up to 72\% likely to improve over baseline RAD.
>
> **Reference:** \
> [1] Rishabh Agarwal, Max Schwarzer, Pablo Samuel Castro, Aaron Courville, and Marc G Bellemare. Deep reinforcement learning at the edge of the statistical precipice. Advances in Neural Information Processing Systems, 2021b.

---

> > ### Author Response · Authors · 2021-11-21
> > **Response to Reviewer FV5x - Part 2**
> >
> >
> > **Fix to style-translate:** \
> > Thank you for pointing out, we have updated the paper to change from "style translate" to "style-translate" (with hyphen).
> >
> > **Adjective before Procgen:** \
> > We have removed the prefix adjective from Procgen throughout the paper. We use the standard evaluation protocol for Procgen. We have updated the description to reflect this in Section 4.1. Thank you for your suggestion.
> >
> > **Typo in related work:** \
> > We have removed "Burda et al 2018" from the list and removed duplicate citations.

---

> > > ### Comment · Reviewer_FV5x · 2021-11-22
> > > **Response**
> > >
> > > Thanks for your detailed replies, and the updates to the paper, I think they have improved it a lot.
> > >
> > > I'll respond to some specific points here:
> > >
> > > > In addition, we have added discussion on your suggested papers except for https://arxiv.org/abs/2106.15587, for which we could not find a peer-reviewed version
> > >
> > > I don't know of a peer-reviewed version, but I also don't think that should be a criteria for whether or not to discuss a paper.
> > >
> > > > Thus our method is orthogonal to many existing algorithms, including strong baseline IDAAC. Combining our method with these algorithms could improve performance; investigating them can be an interesting avenue for future work.
> > >
> > > While I think this is true for most algorithms, it seems not immediately obvious how the method would be combined with other data augmentation techniques such as UCB-DrAC (https://arxiv.org/abs/2006.12862), and so I think the best baseline would be the most performant data augmentation technique, which is not RAD but UCB-DrAC. That would make more sense as a baseline to me. Comparing against that, or justifying how this method could be combined with DA techniques (even if you don't demonstrate it empirically).
> > >
> > > > Mean of the best test results:
> > > It is the best test score of the agent achieved during the entire timesteps. We have modified the Table 1 caption to reflect this.
> > >
> > > In general it is standard to report the mean score at the end of training, as otherwise it is (implicitly) assumed that you always have access to the testing environment to select the model, which isn't a valid assumption.
> > >
> > > > Evaluating the method on another generalization benchmark:
> > > Thank you for your suggestion. We have included the experiments on the distracting control suite in the experiment section (Setup in Section 4.1, results in Section 4.3). Figure 6 shows that our method ARPO performs better compared to baseline PPO on Walker walk environments in both sample efficiency (train) and generalization (test) on two variations of distraction (background and color). Additional results are in Appendix C. [...] With these new results, we now show the improved results of our method on two benchmarks with both discrete and continuous actions spaces with image-based observation. These results demonstrate the usefulness and general applicability of our method.
> > >
> > > I'm glad to see you've evaluated the method on another benchmark, this improves the paper. However, looking at the SAC results from the original DCS paper it seems that your SAC implementation is significantly underperforming on the tasks you've tested it on (in the original paper in the static easy benchmark which is most similar (if slightly harder) than your setting SAC gets 49 and 64 on W-Walk and C-RUn respectively, much lower than your scores.) Further, the C-Run results are so low as to be uninformative of relative performance; none of the policies learned will be effective at all. I also think it woudl be beneficial to do experiments on exactly the benchmark settings layed out in the DCS paper, for reproducibility reasons.
> > >
> > > I this makes me hesitant to accept the paper with just these results. What I think would be most interesting is if the adversarial style transfer approach could be combined with a competitive SAC implementation, as you have combined it with PPO, and see if that improves performance over SAC. This would also demonstrate that this method can be used in both on- and off-policy settings
> > >
> > > > We also show qualitative analysis of the generator's style translation in Appendix E.2.
> > >
> > > Thanks for adding qualitative visualisations, although I think you've overclaiming in their description when you say "We see that the generator retains most of the game semantic while changing the background color and the texture of various essential objects", as the generator does seem to significantly change the visual style.
> > >
> > > # Summary
> > >
> > > Overall, I will raise my score from a 3 to a 5, and my Correctness score from 2 to 3. I think the paper's core idea is interesting but the empirical validation and comparison to baselines is still lacking. I believe work improving the empirical investigation could result in the paper being accepted here although I realise that is unrealistic, so I encourage the authors to continue improving on the idea and submit to another venue in the future, if the paper isn't accepted at ICLR.

---

> > > > ### Author Response · Authors · 2021-11-22
> > > > **Reply to Reviewer FV5x**
> > > >
> > > > **Combine data augmentation with our method (ARPO):** \
> > > > The base RL algorithm (e.g., PPO) in our ARPO method still uses the original observation from the environment while participating in the adversarial objective (see Figure 1). Thus, data augmentation can be applied to the original observations before feeding them to the base RL algorithm to potentially improve the base RL agent, as observed in many data augmentation techniques discussed above.
> > > >
> > > > We have added this discussion in the related work section (Data augmentation in RL).
> > > >
> > > >
> > > > **Comments on qualitative visualizations:** \
> > > > By semantic, we mean the information required for the agent to decide (that is, the true state from observation). For example, in the Walker walk task, changing the color of the robot body should not change the semantic of the observation. However, here style is the color of the robot body, which we want the generator to change. Therefore, it is expected behavior for the generator to change the visual style. Please note that we are not claiming that this is the case for all the translations (e.g., initially generator produces near to random translation, the first row of Figure 16); rather, we describe it for some samples presented in this paper. The purpose of these figures (13, 14, 15, and 16) is to give an intuitive understanding to the reader about our approach.
> > > >
> > > >
> > > > **Implementation of distracting control suite and SAC:** \
> > > > We have used the implementation of distracting control suite available in
> > > > https://github.com/geyang/gym-distracting-control and incorporated it with the RLlib framework for our experiments. This library is built on the original code release (https://github.com/google-research/google-research/tree/master/distracting_control) and makes the environment compatible with the Gym environment (which we find is necessary to use the environment in RLlib framework). We experimented on two distracting settings, dynamic background, and color for easy difficulty mode. We would be happy to incorporate the SAC implementation used to generate results in the distracting control paper (https://arxiv.org/pdf/2101.02722.pdf); however, we could not find an open-source implementation of their methods, including SAC.
> > > >
> > > > **Related work:** \
> > > > We have added a discussion of the paper https://arxiv.org/abs/2106.15587 on the related work (Appendix Section F).
> > > >
> > > > Thank you for revising your score for our paper. Please let us know if you have any additional comments. We appreciate your time.

---

> > > > > ### Comment · Reviewer_FV5x · 2021-11-23
> > > > > **Response**
> > > > >
> > > > > > Combine data augmentation with our method (ARPO):
> > > > >
> > > > > While you're correct that data augmentation could be applied to observations while also being style-transfered for the adversarial objective, most SOTA data augmentation in RL often uses the data augmentation as regularisation (e.g. in DrQ and UCB-DrAC), rather than additional data, and in this case it's less clear how to combine your method with those, and whether the combination would be beneficial: both methods are adjusting the learned representations in different ways, and it's definitely not guaranteed they'd work well together.
> > > > >
> > > > > > Implementation of distracting control suite and SAC:
> > > > >
> > > > > When I said "I also think it woudl be beneficial to do experiments on exactly the benchmark settings layed out in the DCS paper, for reproducibility reasons", I meant that in the DCS paper they define an "easy" benchmark with colour variation, background variation and camera angle variation. I think experimenting on exactly this benchmark would be most beneficial as it would allow direct comparison to their results without having to rerun the experiments. If you did want to rerun the experiments, I believe this is a good implementation of a wide variety of very similar algorithms which could be used: https://github.com/nicklashansen/dmcontrol-generalization-benchmark

---

### Official Review · Reviewer_WXXr · 2021-10-27

**Correctness:** 4
**Technical Novelty And Significance:** 3
**Empirical Novelty And Significance:** 2
**Recommendation:** 8
**Confidence:** 5

**Main Review:**

The general idea of this paper is to train a GAN-like architecture to increase the robustness of RL algorithms to shifts in the input distribution. The Policy Network uses a PPO style loss to train the policy, with the addition of a KL term to ensure that perturbed X’ result in similar actions being taken by the policy. This is a good idea and the authors explain it very clearly in section 3, which I found enjoyable to read. Some more specific comments below:


The KL constraint in equation 3 is the most straightforward way to go about ensuring close observations are mapped continuously into the policy space (in the sense that a small delta in observation space causes a small delta in actions). Did you try a simple MLE loss on the mean of the output distribution? There is always some worry that this sort of KL constraint will lead to mode collapse. But in practice that doesn’t seem to be a problem? Did you consider alternative distributions to parameterize the policy? Some sort of catagorical distribution trick as in DreamerV2? Equation 5 also has a similar KL term. Often, I’ve found that in this kind of setup, it can also help to constrain the policy towards a normal distribution (add a KL(N(0,1), pi) term), to help avoid mode collapse. A similar trick is presented in PEARL.

The clustering pipeline and use of ResNet features was interesting. I think this is a small idea but also novel in my experience.


Is RAD the SOTA data augmentation technique? I’d be interested in seeing comparisons with DRQV2, SODA, SVEA, PAD, and SAC.  What about Learning Task Informed Abstractions, a recent publication on a similar benchmark?

You compare against PPO rather than something like SAC. I suppose that’s fair because this algorithm is based on a PPO style objective. Is on-policy learning truly needed in this case? Would the algorithm not benefit from keeping a buffer as in SAC? I am a bit concerned that this algorithm only marginally improves on PPO in many environments, which is itself a very weak baseline.

Results in table 1 are somewhat hard to parse because the scores are not normalized. It would be ideal if the scores could be normalized, or the results could be presented as a percentage. Maybe in an appendix. I obviously understand the desire to present the raw data.

I think the sample efficiency experiment is important, and I wish more papers included this kind of plot.

The beta value chosen in these experiments is much higher than I would expect. How sensitive is the algorithm to this value? Figure 6 B looks at different values of beta and seems to suggest that this value doesn’t matter very much? Is there any intuition as to why? What happens if beta is set to 0? Is this term even necessary?

In general, the related work section could use some help. There is a rich history of style transfer objectives in inverse reinforcement learning that is missed (AVID: Learning Multi-Stage Tasks via Pixel-Level Translation of Human Videos). This section also runs into the problem of simply listing a lot of different work, but fails to adequately discuss it in the context of the present paper. I’d really like to see this section revised.

**Summary Of The Paper:**

An algorithm is proposed for using a GAN-like objective to generate observation perturbations, thus making RL agents more robust to context shift. This work falls nicely into the recently growing field of data augmentation methods in RL. Results are presented on Procgen.

**Summary Of The Review:**

This method represents an incremental improvement gained by developing a novel architecture. This paper — and the success of the proposed methods — was not surprising. But of course the authors of course should receive credit for getting things to work so well. The quality of results generally ranges from acceptable to quite good, although the quality of ablations falls far behind some comparable papers and can be improved.  Overall, a good paper.

---

> ### Author Response · Authors · 2021-11-21
> **Response to Reviewer WXXr - Part 1**
>
> Thank you for the insightful comments. We have addressed your comments here and updated the paper accordingly. Please let us know if you have any further concerns about this paper.
>
>
> **KL Constraint:** \
> To implement the KL constraint part, we take the mean of the output difference; $KL = (\log \pi(.|x) - \log \pi(.|x')).mean()$.
>
> **Constraining policy pi towards a normal distribution:** \
> Constraining policy pi towards a normal distribution seems like a good idea. However, in our case, the input to the policy pi is an RGB image x (no encoding is applied), and it outputs action distribution. Our understanding is that to compute the KL(N(0,1), pi), we need a probability at value x from the normal distribution N(0,1). Do you have any suggestions on how to get a probability from N(0,1) with an image value?
> Please let us know if you have any further suggestions, and we would be happy to implement and experiment with this idea in our algorithm.
>
> **Comparison with other baselines:** \
> Thank you for the suggestion. Our primary focus is to see how the adversarial style transfer helps learn a robust policy and improves the robustness of the base RL policy used. Thus our method is orthogonal to many existing algorithms and can be potentially combined for better performance. Based on your suggestion, we have added the comparison with SAC on distracting control suite [1] (Appendix Section D). We observe that our algorithm ARPO performs better compared to SAC in various settings.
>
> Just that we do not miss the original version of the papers, could you provide the references of these suggested algorithms: DRQV2, SODA, SVEA, PAD, PEARL, and "Learning Task Informed Abstractions"? We would be happy to incorporate them in the future.
>
> **Regarding on-policy learning and SAC buffer:** \
> We discuss the method of ARPO based on the PPO style objective. However, we think it can be applied to other algorithms as long as we can calculate the KL term for the adversarial training. To the point about the buffer, indeed, we use a buffer to collect the generator training data in our implementation. However, this buffer only filled up initially before GMM clustering and never changed after that. Alternatively, suppose we keep refilling the buffer with the new observations. In that case, we need to train the cluster again from scratch, which will change the generator domain data distribution. Because of this, the generator needs to be trained again from scratch. This process can make the training unstable.
> For this reason, we could not take advantage of the buffer available in the off-policy setup, such as in SAC. However, we could still use the off-policy algorithm as our base RL algorithm. Thus having a separate buffer for the generator allows our method to be incorporated with both on-policy and off-policy algorithms.
>
> **Comparison with SAC:** \
> Based on your suggestion, we have added the comparison with SAC on distracting control suite [1] (Appendix Section D). We observe that our algorithm ARPO (and PPO) performs better than SAC in various settings.
>
> **Additional Experiments on Distracting Control Suite:** \
> To demonstrate the effectiveness of our method, we conduct additional experiments on Distracting control suite [1], which offer continuous control task and is quite different from Procgen. We observe the performance improvement of our ARPO algorithm compared to PPO in several setups. These results are added to the updated version of the paper: Section 4.3, Appendix C, Appendix D.
>
> We have extended our ablation study with new results. We add some qualitative discussion which shows the quality of the generator's output and how it changes over RL training (Appendix E.2).
>
> We think the additional results and discussion can help in understanding the effectiveness of our proposed method.
>
> **Table 1 results presentation and normalized scores:** \
> Thank you for your suggestion. We added a normalized probabilistic measure (suggested recently [2]) on the Procgen generalization results in Appendix B, Figure 8. These are aggregated results overall Procgen environments, and it measures how likely our agent ARPO is to improve over the baseline PPO and RAD. We observe that ARPO is up to 54\% likely to improve over PPO and up to 72\% likely to improve over baseline RAD.
>
>
> **Reference:** \
> [1] Austin  Stone,  Oscar  Ramirez,  Kurt  Konolige,  and  Rico  Jonschkowski.The  distracting  con-trol  suite  –  a  challenging  benchmark  for  reinforcement  learning  from  pixels.arXiv  preprintarXiv:2101.02722, 2021
>
> [2] Rishabh Agarwal, Max Schwarzer, Pablo Samuel Castro, Aaron Courville, and Marc G Bellemare.Deep reinforcement learning at the edge of the statistical precipice.Advances in Neural Informa-tion Processing Systems, 2021b.

---

> > ### Author Response · Authors · 2021-11-21
> > **Response to Reviewer WXXr - Part 2**
> >
> >
> > **Choosing $\beta$ value:** \
> > We observed some slight variation in the train test score for different beta values (in the ablation study), and overall we found the algorithm is less sensitive to it.
> > If we set the beta value to 0, then the adversarial min-max KL objective (and the style transfer generator) will not be part of the algorithm, and our algorithm will fall back to the base (PPO in our setup) algorithm. Thus, these beta values control how much the adversarial objective influences RL training. The magnitude of the beta values also depends on the implementation of the loss function, especially the scale and range of the resulting loss values. Thus depending on different RL tasks, we suggest these values be tuned accordingly.
> >
> > **Related work:** \
> > We have revised the related work section based on your suggestion. Please refer to the related work section (now in the Appendix Section F, because of the space limitation).

---

> > > ### Comment · Reviewer_WXXr · 2021-11-22
> > > **Satisfied with the improvements**
> > >
> > > I have reviewed the improvements to the paper, and I am now quite satisfied. While I do wish there were more baseline comparisons such as UCB-DrAC and the benchmarks from DCS, I think there has been a reasonable effort made by the authors especially with the inclusion of SAC as a baseline. I would like to see a more competitive SAC baseline implemented for the camera ready version of this paper, as mentioned by other reviewers. However, I think these results are already interesting, and it seems clear to me at least that the author's proposed method is both novel and helpful at obtaining stronger generalization results. I think the baseline and experimental results clear the bar and are comparable to recent publications in this field.

---

### Official Review · Reviewer_eK4b · 2021-11-01

**Correctness:** 4
**Technical Novelty And Significance:** 3
**Empirical Novelty And Significance:** 3
**Recommendation:** 6
**Confidence:** 3

**Main Review:**

To learn robust features for decision-making, the generator of the StarGAN is encouraged to style transfer the state space such that the action distribution under the new transferred state will be distinct in terms of KL divergence with the original policy. The generator is further trained with a cyclic consistency loss.

The domain discriminator tries to tell about different domains.

Using Proximal Policy Optimization (PPO) as an example, the policy network is trained with an augmented loss of minimizing the KL divergence of the action distribution between original and style-transfered state.


Experiments are done on the Procgen environments in comparison to PPO and RAD(data augmentation). Ablation study is done with respect to different linear coefficients, combining the different losses in the adversarial learning.

# Strong points
- The paper integrate style-transfer into the training of reinforcement learning algorithms to enhance the algorithm robustness against spurious features.
- The experimental setting is explained relatively in detail. Experimental results look promising.

#Issues
1. In section 3.3, GMM is only used in the beginning to generate different clusters for usage in Star-GAN learning. Will this be sensitive to initial sample distribution?
2. Could you elaborate on how the GMM clustering is done? Simply pool all state observations and do the clustering? e.g. for a tuple s,a,s', s and s' are treated  equally?

## Minor issues
1. 3rd paragraph in section 1, there is a citation on Anonymous.

# Questions
1. Could you explain the benefit of the gradient penalty term in Eqn. (6)?


**Summary Of The Paper:**


To achieve more robust behavior of reinforcement learning algorithms, in the deep learning part,  confounding features need to be removed from the decision-making process. To achieve this, the paper proposed first clustering the state space into $n$ clusters/domains and use StarGAN to style-transfer from one cluster to another.

**Summary Of The Review:**

I like the innovative point that policy is not only learned to optimize return but also robustness against style transfer of state observation. Compared to data augmentation methods, the proposed methods showed promising methodology and empirical advantage.
Issues in the above section need to be addressed however.
.

---

> ### Author Response · Authors · 2021-11-21
> **Response to Reviewer eK4b**
>
> Thank you for your helpful comments. We have addressed your comments here and updated the paper accordingly. Please let us know if you have any further concerns about this paper.
>
> **GMM training and initial sample distribution:** \
> For stable generator training, we keep a fixed-size buffer to store observations for GMM training and then use it for generator training. We only apply GMM at the beginning, when the fixed buffer is full. If we change the clustering later, it potentially changes the distribution of data in different clusters, requiring the generator to be trained from the beginning.
> Yes, this setup might be sensitive to the initial sample distribution. Ideally, this initial dataset should be a good representative of the entire training observations of the environment. However, due to the RL training nature, the initial policy usually behaves randomly as no learning is being done, which might help collect a diverse dataset. Therefore, we suggest setting the dataset size depending on the complexity of the observation space of the environment. In our experiments, for Procgen, we set the dataset size to 1000, and for the distracting control (added to the updated paper), the dataset size is 2000.
>
> **Elaboration on GMM clustering:** \
> Yes. We store all the state observations from the agent's interaction with the environment in a buffer and then train GMM clustering when the buffer is full. We treat s and s' equally.
>
> **Anonymous citation:** \
> Unfortunately, we have to keep the author name of this paper anonymous due to the anonymity policy of the venue where it is submitted. We will update this when the anonymity is lifted.
>
> **Benefit of the gradient penalty term in Eqn. (6)** \
> The gradient penalty in Eqn. (6) helps to stabilize the GAN training process and generate high-quality images [1]. This method has been shown to perform better than traditional GAN training [2, 3]. We have added a discussion about this under Eqn. (6) to reflect this.
>
> **Reference:** \
> [1] Yunjey Choi, Minje Choi, Munyoung Kim, Jung-Woo Ha, Sunghun Kim, and Jaegul Choo.  Star-gan:  Unified generative adversarial networks for multi-domain image-to-image translation. In Proceedings of the IEEE conference on computer vision and pattern recognition, pp. 8789–8797,2018.
>
> [2] Martin Arjovsky, Soumith Chintala, and Leon Bottou. Wasserstein generative adversarial networks. In International conference on machine learning, pp. 214–223. PMLR, 2017.
>
> [3] Ishaan Gulrajani, Faruk Ahmed, Martin Arjovsky, Vincent Dumoulin, and Aaron Courville.   Im-proved training of wasserstein gans.arXiv preprint arXiv:1704.00028, 2017.

---

### Official Review · Reviewer_f6jT · 2021-11-03

**Correctness:** 3
**Technical Novelty And Significance:** 2
**Empirical Novelty And Significance:** 2
**Recommendation:** 5
**Confidence:** 3

**Main Review:**

Strength
- The idea of combining style transfer and RL to improve generalization is reasonable. As far as I know, the idea is novel in this setting.

Weakness
- There are multiple overclaim in this paper. 1) In the abstract, there is no empirical
nor theoretical evidence that ARPO finds optimal policy. It is highly skeptical that
ARPO can find an optimal policy in practice as the added KL regularization will rarely converge to zero.
- It might be better to discuss whether the policy will converge to the optimal policy, where KL term in eq 3 reaches zero at the end.
- An important ablation study could be showing how does the KL regularization change during the
training.
- The three run averages of results present large variance, and it is not very convincing to me ARPO
really achieves statistically better generalization results. Based on the results in Table 1,
there is no significant improvement from ARPO compared to baselines.
- There is no discussion nor ablations on whether cycle consistency loss finds the
underlying structure of MDP. It would be better at least to provide examples of style-transferred images compared with original images.

**Summary Of The Paper:**

This work proposes Adversarial Robust Policy Optimization (ARPO) that trains a generator to transfer the observation style so that policy taking different actions, the policy optimizes the long-term reward plus a regularization that minimizes the KL divergence between policy outputs given style-transferred image and original image. In this way, ARPO tries to learn underlying semantic representations for better generalization and being robust to the perturbations. The experimental results on some Procgen envs show improved results compared to PPO and RAD.

**Summary Of The Review:**

This paper proposes a novel idea that combines style transfer with RL to improve the
generalization performance. The experimental results show the proposed approach is competitive with the existing baselines. The results could be better improved by adding more runs and more ablations that approved the effectiveness of ARPO. More theoretical studies are needed to understand whether the algorithm converges to optimal policy.

---

> ### Author Response · Authors · 2021-11-21
> **Response to Reviewer f6jT**
>
> Thank you for your helpful comments. We have addressed your comments here and updated the paper accordingly. Please let us know if you have any further concerns about this paper.
>
> **Paper claims, optimality, and KL value:** \
> The adversarial min-max KL component $KL(\pi_\theta(.|x_t), \pi_\theta(.|x_t'))$ become zero when the RL policy is optimal ($\pi^*$) and the generators only perturbs (translates) the irrelevant part of all observations. In that case, the optimal policy only focuses on the relevant part of the observations, that is, the true state; thus, any changes in irrelevant part due to style transfer will be ignored. At that point the KL component become zero as the $\pi^*_\theta(.|x_t) = \pi^*_\theta(.|x_t')$. The KL value can also be zero when both policy and generator are random, at the beginning of the training. We confirm this observation empirically in Figure 17.
>
> Note that in practice, the algorithm is usually trained for limited timesteps, and thus the adversarial training might not converge to optimal. However, empirically we observe that our algorithm ARPO achieves improved performance in many challenging Procgen and Distracting control suite tasks.
> We added this discussion in the paper at the end of Section 3.
>
> **Ablation study on KL regularization:** \
> Thank you for the suggestion. We have added the KL regularization ablation study in Appendix Section E.3 (Figure 17). At the beginning of the training, both the policy and generator behave randomly; thus, the KL values become nearly zero. Gradually, the KL started to increase as the policy learning progressed with the adversarial generated observation. Eventually, the policy tries to adjust to the changes in the observation, and the KL values become stable. However, the stability cutoff time (policy iteration) of each environment is different, suggesting that the KL regularization behavior might be different depending on the environment complexity. Ideally, with enough training, the KL values should converge to zero (again) if the policy reaches optimal (for true MDP state) and the generator fully recovers the true state from the observation and changes all the irrelevant information from the observation. Note that we limit the policy to train until a cutoff time (22 Million RL timesteps).
>
> **Number of runs and more experiments:** \
> For the Procgen benchmark experiments, we follow the benchmark experimental setup where we run each experiment for 3 random seeds. We include additional results on Distracting control suite [1] in the updated version, where our algorithm ARPO achieves better results than the baseline in various settings (Figure 6). Furthermore, we added aggregated results on all 16 Procgen environments in Figure 8 (Appendix). This result leverages a recently proposed probabilistic measure [2] of our agent ARPO is to improve over the baseline PPO and RAD. We observe that ARPO is up to 54\% likely to improve over PPO and up to 72\% likely to improve over baseline RAD in the Procgen generalization setup.
>
> **Ablation on cycle consistency and examples of style-transferred images:** \
> Thank you for the suggestion. In the updated paper, we added qualitative discussion on the style-transferred images compared to the original. The results are in Figures 13, 14, 15, and 16. We observe that in many cases, the generator can translate images while keeping the task-related semantic same. Interestingly, we observe that the generator blackout some irrelevant background.
> Furthermore, Figure 16 shows how the generator evolves over the RL training phase. Please see the Appendix and Figure caption for more discussion. These results might indicate that the cycle consistency loss and the adversarial KL regularization help the agent find the underlying MDP's true state from the observation.
>
> **Reference:** \
> [1] Austin Stone, Oscar Ramirez, Kurt Konolige, and Rico Jonschkowski. The distracting control suite – a challenging benchmark for reinforcement learning from pixels. arXiv preprint arXiv:2101.02722, 2021.
>
> [2] Rishabh Agarwal, Max Schwarzer, Pablo Samuel Castro, Aaron Courville, and Marc G Bellemare. Deep reinforcement learning at the edge of the statistical precipice. Advances in Neural Information Processing Systems, 2021b.

---

> > ### Comment · Reviewer_f6jT · 2021-11-22
> > **Reply**
> >
> > On the optimality. The results in Fig 17 conflict with the claims in the abstract and main text: "Adversarial Robust Policy Optimization (ARPO), to find an optimal policy that generalizes to unseen environments. "  The objective of ARPO preserves optimality is different from finding the optimal policy via ARPO. Empirically, it doesn't find optimal policy. Theoretically, I haven't see any rigorous investigation on why ARPO can find an optimal policy.

---

> > > ### Author Response · Authors · 2021-11-22
> > > **Reply to Reviewer f6jT**
> > >
> > > Thank you for pointing it out. We have modified the language regarding describing our algorithm in the abstract and main body. We mainly replace the term "optimal" with "robust" which we think best describes our ARPO method. Please let us know if you have any additional feedback.

---

### Author Response · Authors · 2021-11-21
**To all reviewers**

We appreciate your time in reviewing our paper and thank you for your valuable comments. We have taken your suggestions into account and revised our paper substantially. We address each reviewer's comments individually under their reviews, and here is a summary of changes.

- Included experiments on a new generalization benchmark that is on distracting control suite.
- Revised paper claims based on the reviewers' suggestions.
- Improved the ablation study with new results, including KL regularization, example style-translations (Procgen and distracting control), and qualitative analysis of generator training.
- Revised related work incorporating reviewers' suggestions.
- Modify representation of results in Table 1 and add a normalized score metric to report Procgen results.
- Added discussion on limitations about our proposed method.

Please see the individual comment for a detailed response to each of your concerns. Please let us know if anything is unclear and if you have any additional feedback about our paper. Again, thank you for your comments, and we appreciate your time.

---

### Decision · Program_Chairs · 2022-01-20

**Decision:**

Reject

**Comment:**

I thank the authors for their submission and active participation in the discussions. This papers is borderline with reviewers WXXr and eK4b leaning towards acceptance and reviewers f6jT and FV5x leaning towards rejection. On the positive side, reviewers remarked that the paper is interesting [FV5x] and novel [FV5x,f6jT,eK4b,WXXr]. However, there all reviewers found some flaws with respect to the execution and empirical validation [FV5x], specifically around lacking baselines [FV5x,WXXr] and some ablations [f6jT,WXXr]. I side with the comment made by reviewers FV5x as well as WXXr that a comparison to stronger baselines (UCB-DrAC) is warranted. Therefore, I recommend that this paper is not ready for publication at this point and that it will benefit greatly from another iteration with stronger empirical results. I want to very strongly encourage the authors to further improve their paper based on the reviewer feedback.